# Temperature Control of Yellow Photoluminescence from SiO_2_-Coated ZnO Nanocrystals

**DOI:** 10.3390/nano12193368

**Published:** 2022-09-27

**Authors:** Narender Kumar, Vijo Poulose, Youssef Taiser Laz, Falguni Chandra, Salma Abubakar, Abdalla S. Abdelhamid, Ahmed Alzamly, Na’il Saleh

**Affiliations:** 1Department of Chemistry, College of Science, United Arab Emirates University, Al Ain P.O. Box 15551, United Arab Emirates; 2Academic Support Department, Abu Dhabi Polytechnic, Al Ain P.O. Box 15551, United Arab Emirate; 3Science Division, New York University Abu Dhabi (NYUAD), Saadiyat Island P.O. Box 129188, United Arab Emirates

**Keywords:** ZnO nanocrystals, temperature control, time-resolved photoluminescence, nonradiative relaxation

## Abstract

In this study, we aimed to elucidate the effects of temperature on the photoluminescence from ZnO–SiO_2_ nanocomposite and to describe the preparation of SiO_2_-coated ZnO nanocrystals using a chemical precipitation method, as confirmed by Fourier transform infrared (FTIR) and powder X-ray diffraction analysis (XRD) techniques. Analyses using high-resolution transmission microscopy (TEM), energy-dispersive X-ray spectroscopy (EDX), dynamic light scattering (DLS), and electrophoretic light scattering (ELS) techniques showed that the new nanocomposite has an average size of 70 nm and 90% silica. Diffuse reflectance spectroscopy (DRS), photoluminescence (PL), and photoluminescence-excitation (PLE) measurements at different temperatures revealed two emission bands at 385 and 590 nm when the nanomaterials were excited at 325 nm. The UV and yellow emission bands were attributed to the radiative recombination and surface defects. The variable-temperature, time-resolved photoluminescence (VT-TRPL) measurements in the presence of SiO_2_ revealed the increase in the exciton lifetime values and the interplay of the thermally induced nonradiative recombination transfer of the excited-state population of the yellow emission via deep centers (DC). The results pave the way for more applications in photocatalysis and biomedical technology.

## 1. Introduction

In recent years, zinc oxide (ZnO) has gained significant attention because of its promising applications in photonics [1], spintronics [2], and optoelectronic devices such as LEDs (light emitting diodes) [3], phase-change memory devices [4], solar cells [5], sensors [6], antibacterial activity [7], plant growth [8], electrochemistry [9], and photocatalysts [10]. 

To prepare photoluminescent nanocomposites from ZnO, because ZnO alone tends to aggregate, researchers have used several techniques such as annealing, double-jet precipitation, spray-drying, and sol–gel methods to prepare ZnO–SiO_2_ nanocomposites with pure UV emissions [11] at 360 and 400 nm [12], multicolored emission (from 460 to 550 nm) depending on the aging time [13], infrared emission via energy transfer to Yb^3+^ [14], and multicolored emission from 365 to 517 nm depending on the calcination temperature [15]; all nanocomposites have been excited at 325 nm. Recently, due to the interesting photoluminescence (PL) properties of ZnO–SiO_2_ films, many researchers have used nanomaterials as antidotes in human dermal fibroblast cells [16], as antibacterial agents [17], and as photocatalysts, which are one of the most popular applications [18].

The dependence of the photocatalytic activity of ZnO/SiO_2_ nanomaterials on temperature [18,19] needs further elucidation. For instance, the temperature-dependent PL of ZnO/SiO_2_ nanomaterials has never been investigated. Time-resolved photoluminescence (TRPL) spectroscopy is one the most important tools to investigate the bandgap, defect, and impurity of any nanostructure or bulk material. The UV and visible PL of ZnO–SiO_2_ films have already been attributed to radiative recombination and surface defects [11]. Here, we analyze TRPL measurements at different temperatures to understand the photophysical role of SiO_2_ coating in generating nanocomposites with temperature-dependent PL aside from its role in increasing the exciton lifetime values [20]. The interplay of thermally induced nonradiative recombination transfer via deep centers (DC) of the excited-state population of the yellow emission band is confirmed.

## 2. Experimental Methods

### 2.1. Synthesis of Zinc Oxide Nanocrystals

First, the ZnO nanoparticles were generated from a colloidal solution of ZnO in 250 mL ethanol before mixing them with 5.5 g of zinc acetate dehydrate (purity > 98% by Sigma-Aldrich, Saint Louis, MO, USA). Then, the final suspension was homogenized by heating under reflux for 1 h [21]. The distillation method was used to remove 150 mL of the solvent. After adding a new fresh volume of ethanol, 1.39 g of lithium hydroxide monohydrate (Aldrich) was added, and the new suspension was blended in an ultrasonic bath at 333 K. The dispersed mixture for 1 h was assumed to contain pure nanoparticles of ZnO, and then the mixture was filtered through a 0.1 mm membrane filter to remove undissolved LiOH. Then, the final white precipitate of the ZnO nanoparticles was heated in 100 mL of 10% deionized water in ethanol at 333 K, centrifuged, and then washed four times with an ethanol/water mixture (19:1) to remove residual ionic compounds [22].

### 2.2. Coating of ZnO Nanocrystals with SiO_2_

An amount of 0.5 g of ZnO was dispersed with silica (90 wt%), and then the resultant suspension was sonicated for 5 min, stirred overnight in LiOH by centrifugation, and washed four times with ethanol/water mixture (19:1) to remove residual ionic compounds, and then dried in a vacuum oven at 333 K.

### 2.3. Fourier Transform Infrared (FTIR) Spectroscopy

Identification of the functional group was performed using a Thermo Scientific Nicolet instrument, and all samples were prepared using FTIR grade KBr which was sourced from Sigma-Aldrich. The samples were dispersed in water and physically scratched on a glass plate. The resulting water suspension was dried by evaporating water with the aid of a hot plate and an oven at 383 K, and then KBr was weighed quantitatively at a ratio of 1:100 (wt/wt).

### 2.4. High-Resolution Transmission Microscopy (HR-TEM)

The internal morphology (final shape and length) of the SiO_2_-coated ZnO slides was deduced from the TEM images which were taken using an FEI-Titan 300 electron microscope operating at 200 kV (HR-TEM, Hillsboro, OR, USA).

### 2.5. Energy-Dispersive X-ray Spectroscopy (EDX)

The elemental analysis was conducted using EDX and HR-TEM.

### 2.6. Powder X-ray Diffraction Analysis (XRD)

The powder X-ray diffraction analysis was performed using a Rigaku MiniFlex X-ray diffractometer (Japan, https://www.rigakuedxrf.com). The diffractogram recorded at 2θ, from 3° to 90°, and the instrument parameters were: target metal, Cu; filter, Kα; voltage, 45 kV; and current, 40 mA.

### 2.7. Size Distribution and Zeta Potential Analysis

The particle size (z-average hydrodynamic diameter (nm)) and zeta potential (ζ) were measured by dynamic light scattering (DLS) and electrophoretic light scattering (ELS) performed on a Malvern zetasizer Nano ZS instrument (Malvern Instruments, UK). All measurements were performed at 25 °C after 120 s of temperature equilibration at a backscattering angle of 173°. Using the Stokes–Einstein equation, the z-average hydrodynamic diameter (nm) was calculated from the diffusion coefficient. The zeta potential was calculated from the electrophoretic mobility using the Smoluchowski assumption. The z-average (nm) and ζ (mV) were reported as the mean of three replicates and standard deviation.

### 2.8. Diffuse Reflectance Spectroscopy (DRS)

The absorption spectra of the SiO_2_-coated ZnO slides were obtained by using the Kubelka–Munk conversion (K–M = (1 − R)^2^/2R) of the recorded DRS spectra at room temperature for the solid samples on an FS5 spectrometer (Edinburgh, UK) equipped with a SC-30 (integrating sphere) as the sample holder. The specular reflection of the sample surface light was removed from the signal by directing the incident light at the sample at an angle of 0°; only the diffusive reflected light was measured. Polytetrafluoroethylene (PTFE) polymer was used as the reference. From the DRS spectra, the bandgap energy (E_g_) values of the solid samples were calculated using E_g_ = 1240 eV nm L^−1^, where l is the absorption edge (in nm).

### 2.9. Photoluminescence (PL) and Photoluminescence-Excitation (PLE) Measurements

The PL spectra of the SiO_2_-coated ZnO slides were recorded using an FS5 spectrometer (Edinburgh, UK) equipped with an SC-10 as the sample holder with a front-face geometry, which also utilized a Xenon lamp to excite the sample at 320 and 375 nm with long pass filters (LPFs) at 375 (or 395 nm) and 420 nm, respectively, which were placed between the sample holder and the emission monochromator. The excitation spectra were collected at 411 and 550 nm. All measurements were performed at two different temperatures, i.e., 263 and 298 K.

### 2.10. Variable-Temperature (VT) Measurements

For the temperature variation experiments, a thermoelectrically cooled four-window cuvette holder was used together with an instrument software-powered controller to enable stable control of sample temperatures, from 263 to 378 K.

### 2.11. Excited-State PL Lifetime Measurements and Time-Resolved Photoluminescence (TRPL) Measurements

The emission decays were measured by using a time-correlated single-photon counting (TCSPC)-based Edinburgh instrument (LifeSpec II spectrometer, Edinburgh, UK). The source used was either a picosecond diode laser with an λ_e__x_ of 375 nm with an instrument function of ~30 ps or a laser diode at 320 nm with an instrument function of ~90 ps and a specific LPF as described in the text; both sources were set at a repetition rate of 5 MHz. The PL decays were collected with a total count rate of 10,000 counts/s by using a red-sensitive high-speed PMT detector (H5773-04, Hamamatsu, Japan). The data were globally fitted to a tri-exponential model function convoluted with the appropriate instrument response function (IRF), utilizing a least squares statistical analysis (chi-square and residual plot were used to assess the goodness of fit). The average PL lifetime ($\bar{\tau }$) was calculated using $\bar{\tau }\underset{i}{\sum }f_{i}\tau_{i}$), where *f_i_* is the extracted fractional intensity contribution. The estimated experimental error was 2% for PL lifetime less than 1 ns and 20% for PL lifetime between 20 and 100 ns.

### 2.12. DAS Measurements

For more adequate fitting, the emission decays collected every 10 nm over the entire emission spectra of SiO_2_-coated ZnO slides with a dwell time of 10 s at each wavelength were globally fitted to a tri-exponential model function, and then convoluted with an instrument response function (IRF) of ~30 ps. The time-resolved data were specifically analyzed using the Edinburgh FAST software, in which decay-associated spectra (DAS) were constructed from the extracted intensity-contribution fraction (*f_i_*) calculated from the pre-exponential amplitudes (*B_i_*), as follows:(1)$$I\left(t\right)=\underset{i}{\sum }B_{i}exp\left(-t/\tau_{i}\right)$$
(2)$$f_{i}=\frac{B_{i}\tau_{i}}{\sum_{j}B_{j}\tau_{j}}$$

## 3. Results and Discussion

### 3.1. Material Characterizations of SiO_2_-Coated ZnO Nanocrystals (Elements, Shape, Crystallography, Size Distribution, and Charges)

Appendix A shows the FTIR spectrum for the prepared SiO_2_-coated ZnO slides as compared with those spectra for standard SiO_2_ and ZnO powders as control samples. The characteristic bands at 3395 cm^−1^ (ZnO), 3430 cm^−1^ (SiO_2_), and 3448 cm^−1^ (sample) are attributed to the stretching vibration of intermolecular hydrogen bonds that exist between the adsorbed water molecules. The characteristic bands in ZnO at 1380 cm^−1^ shifted in the SiO_2_-coated ZnO nanocrystals to 1385 cm^−1^, whereas the characteristic Si-O-Si asymmetry at 1097 cm^−1^ observed in SiO_2_ powder shifted to a new value at 1025 cm^−1^. Additionally, the characteristic bands in SiO_2_ at 1643 cm^−1^ also shifted to 1596 cm^−1^.

The surface morphologies of the SiO_2_-coated ZnO nanocrystals were examined using transmission electron microscopy (TEM). Figure 1 shows a TEM image of the SiO_2_-coated ZnO nanocrystals. The ZnO nanocrystals have a dense shell of SiO_2_ coating their surface, with a thickness of about 75 nanometers, as measured by the TEM imaging. This size corresponds to that calculated from the XRD patterns and measured by DLS. To prove the presence of SiO_2_, the elemental analysis of SiO_2_-coated ZnO nanocrystals was carried out using energy dispersive spectroscopy (EDX). The EDX spectrum confirmed the presence of elements Zn, O, and Si in the prepared sample (Table 1). The carbon peak is due to the supporting substrate while collecting the TEM images or from the methanol solvent used in the preparation. Moreover, since the SiO_2_-coated ZnO has 90 wt% silica, it is reflected in the higher atomic and weight percentages of Si as compared with Zn.

The analyzed sample as a thin film on top of a glass plate under room temperature displays peaks less intense, but the diffraction patterns are specific and comparable (Appendix A). The diffraction pattern of ZnO shows distinct and sharp peaks at 2θ values of 31.68°, 34.14°, 35.87°, 47.47°, 56.60°, 62.79°, and 67.92°. The broad peak at 2θ of around 20–30 corresponds to the amorphous silica. The diffraction pattern of the sample shows a unique pattern at 2θ values of 35.62°, 37.40°, 48.26°, 57.49°, 64.10°, and 68.95°.

The average size measured for the sample is 51.22 d. nm with a polydispersity index of 0.206. The zeta potential (ζ) value measured for the sample is −30.8 (Appendix A).

### 3.2. Optical Properties of the SiO_2_-Coated ZnO Nanocrystals

By increasing the temperature of the SiO_2_-coated ZnO slides from 263 K to 298 K, only the UV–visible absorption edge shifts from 335 to 355 nm, which corresponds to decreasing its optical bandgap from 3.7 to 3.5 eV (Figure 2), whereas the optical bandgap remains unchanged at 3.3 eV (375 nm).

The measured photoluminescence (PL) and excitation spectra (PLE) of the SiO_2_-coated ZnO samples (Appendix A) confirm the presence of two electronic species [11,12,13,14,15], one emits at 385 nm (UV band), whose excitation spectrum has a maximum at 323 nm, and the other emits at 590 nm (yellow band) with an excitation peak position shifted from 323 to 348 nm as a shoulder. The results were confirmed by selective excitation of the yellow-emitted species at 375 nm instead of 320 nm, noting a higher intensity which was in agreement with the literature on similar nanocomposites [13]. Notably, the peak at 385 nm appeared at 400 nm because of the use of a long pass filter at 375 nm.

The measured PL spectra upon excitation at 320 nm of the ZnO/SiO_2_ solid samples as a function of temperatures (Experimental Methods) reveal that, with an increase in the temperature from 263 to 378 K, the PL intensity decreases by one order of magnitude with a concomitant blue shift in the yellow emission peak position from about 590 to 575 nm (Figure 3A), while the UV peak at 385 nm (appeared at 411 nm when an LPF at 395 nm was used) remains unchanged. The results are confirmed by the selective excitation of the yellow band at 375 nm (Figure 3B). The data, generated in multiple cycles, confirm nearly 100% restoration of the spectra with changes in temperature from 298 to 378 K (Figure 3C,D).

The emission decays were measured to better comprehend the nature of the fluorescent species in the solid state with decreasing temperature. The estimated excited-state PL lifetimes monitored at 411 and 541 nm when the samples were excited at 320 nm, respectively, at 298 K and 378 K, are listed in Table 2 and displayed in Figure 4. All emission traces were fitted to tri-exponential model functions after being convoluted with IRF (Experimental Methods).

For comparison purposes, the estimated excited-state PL lifetimes from the decays monitored at 541 nm alone (Appendix A) when the yellow-emitted species were selectively excited at 375 are compiled in Appendix A.

An analysis of the emission decays in Table 2 reveals no change in the average excited-state PL lifetime value (~33 ns) at 411 nm. In contrast, the average excited-state PL lifetime decreases at 541 nm (viz. from 66 to 24 ns, Table 2) with an increase in temperature from 298 to 378 K, indicating the generation of a new nonradiative channel, see below. Similar observations were confirmed upon the selective excitation of the 541 nm species (exciton) at 375 nm (viz. from 100 to 59 ns in Appendix A).

The DAS measurements more accurately indicate the effects of temperature on both the PL lifetime values and the contribution values of all components conducted globally at all wavelengths. It also helps us to assign each lifetime component to its DAS maximum (Table 3 and Appendix A). It is observed that the PL lifetime values and amplitudes of the first two shorter Lifetimes **1** and **2** are not affected by temperature change (Table 3). However, the third component is solely affected by temperature. The decreased PL lifetime value (Table 3) and the decreased contribution of the third lifetime Component **3** (from 40 to 6%) confirm its population transfer through a nonradiative process, see below.

The origin of UV and visible emission of ZnO has been attributed to near band edge emission and intrinsic defects such as oxygen vacancies, respectively [11]. One report suggested that the decrease in the intensity of visible emission at 298 K has resulted when surface defects are effectively passivated by the SiO_2_ [11]. Another research work reveals an increase in the exciton lifetime values in the presence of SiO_2_ [20]. In the present work, we specifically observed temperature-dependent changes in the intensity of the visible emission for the first time in the presence of SiO_2_.

The results also agree with DRS measurements at different temperatures in which the bandgap at 3.3 eV belongs to ZnO [1], while the bandgap at 3.7 eV, which is sensitive to temperature, is associated with the SiO_2_-coated ZnO nanocrystals.

The % of Component **3** is calculated using the relation $\%3=\frac{\;DAS\;amplitudefor3}{DAS\;amplitudes\;for1+2+3}$. For the time-resolved fluorescence measurements, the time resolution was ~30 picoseconds and the excitation wavelength was 375 nm.

### 3.3. The Origin of Temperature-Dependent PL of ZnO/SiO_2_ Nanomaterials

PL temperature control of ZnO has already been reported by several researchers. Peng et al. studied the temperature-dependent photoluminescence properties of needle-like ZnO nanostructures and observed that the intensities of all transitions occurred in near bandgap and visible regions and decreased with an increase in temperature along with the excitation power [23]. Moreover, Li et al. reported on exciton localization and stimulated emissions of ZnO nanocrystals and observed very weak visible emissions at low temperatures [24]. An inverse relation between PL with an increase in temperature was reported by Mhlongo et al. [25]. Bouzouraa et al. analyzed the surface property of ZnO and found that it was highly dependent on temperature [26]. It was later found that the morphological and optical properties of ZnO nanocrystals could further be tuned for optoelectronic device requirements by changing the processing temperature [27]. In another report on ZnO nanocrystals, it was observed that the temperature-dependent behavior of exciton transitions and oxygen vacancy recombinations [28] depended on temperature and decreased with an enhancement in the yellow band due to increasingly prominent localized bound exciton emissions.

Temperature control of PL from ZnO material upon coating by SiO_2_ reveals, for the first time, the generation of thermally controlled nonradiative channel in ZnO-based nanocomposites. In general, nonradiative recombination processes that compete with radiative donor-acceptor recombination pair (DAP) emission from the emitting centers in solids can occur through either multiphoton emission (MPE) or energy transfer (Auger) processes [29]. In our ZnO-SiO_2_ system, we confirm, for the first time, that the recombination of the free carriers via deep centers (DC) competes with the surface defect-assisted yellow emission. This is true because the capture process via DC is thermally activated as opposed to other nonradiative mechanisms [29].

## 4. Conclusions

In this study, we aimed at elucidating the effects of temperature on the photoluminescence from ZnO–SiO_2_ nanocomposites. The formation of SiO_2_-coated ZnO nanocrystals is confirmed by using several techniques such as FTIR, TEM, EDX, XRD, DLS, zeta potential, PL, PLE, TRPL, and the results elucidate the fundamental photophysical mechanism behind the efficient reversible switching of the yellow emission properties of SiO_2_-coated ZnO nanocrystals by changing the ambient temperatures. Two emission bands at 385 and 590 nm are observed when the nanomaterials are excited at 325 nm. The UV and yellow bands are attributed to the radiative recombination (3.3 eV, ZnO) and surface defects (3.7 eV, ZnO–SiO_2_). We confirm the contribution of the DC nonradiative recombination mechanism, for the first time, as an explanation for the decrease in yellow emission by increasing the temperature. The results pave the way for implementing these nanocomposites in the development of reversible optoelectronic devices with temperature control and for temperature-dependent photocatalytic applications.

## Figures and Tables

**Figure 1 nanomaterials-12-03368-f001:**
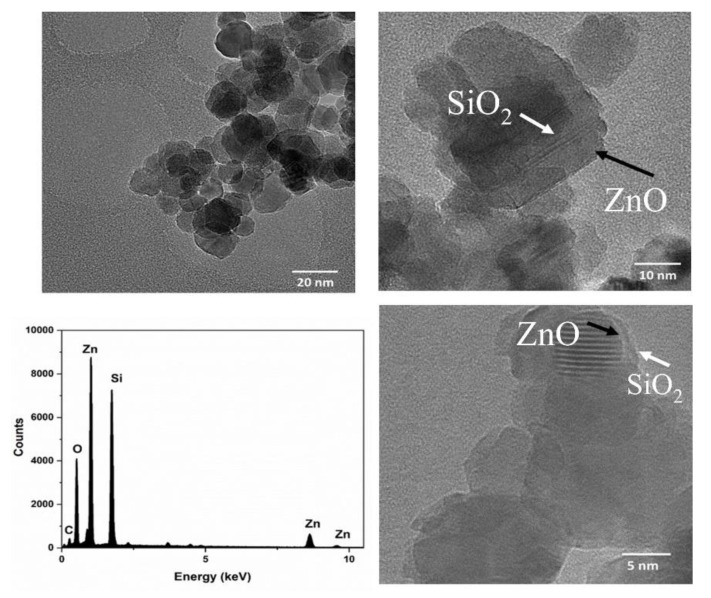
TEM images and the corresponding EDX spectrum of the SiO_2_-coated ZnO nanocrystals.

**Figure 2 nanomaterials-12-03368-f002:**
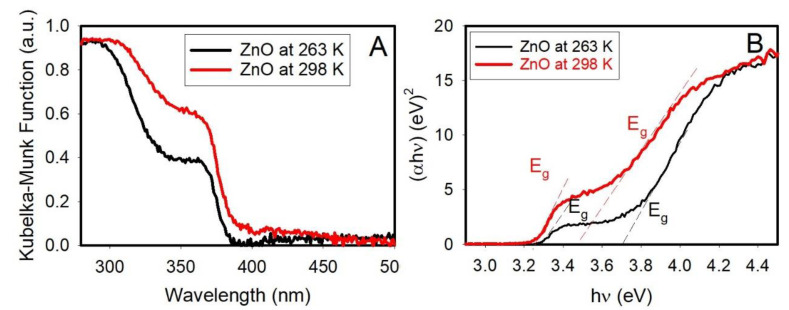
Diffusive reflectance spectra (DRS) (**A**); and the Tauc plots (**B**) of SiO_2_-coated ZnO slides at 263 and 298 K (ambient temperatures) suggest direct optical bandgaps.

**Figure 3 nanomaterials-12-03368-f003:**
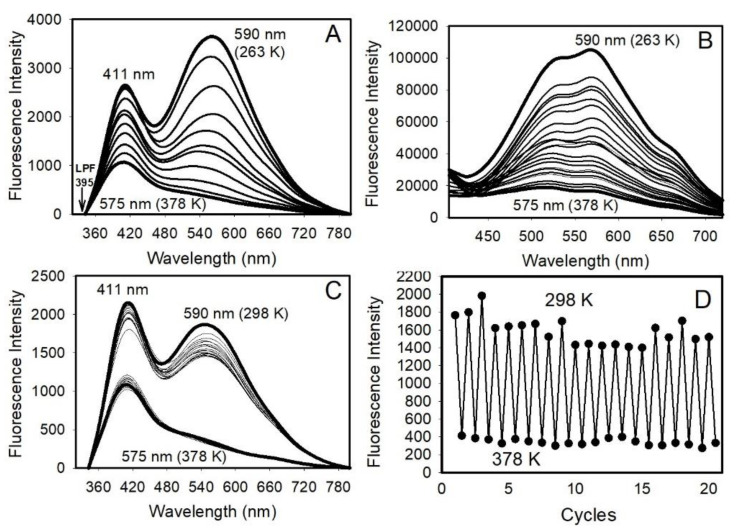
Photoluminescence (PL) spectra upon excitation at 320 nm (**A**) and 375 nm (**B**) at different temperatures from 263 to 378 K; (**C**) the data generated in several cycles from 298 to 378 K; (**D**) the corresponding plot monitored at 590 nm. Notably, the peak at 385 nm appeared at 411 nm because of the use of a long pass filter at 395 nm.

**Figure 4 nanomaterials-12-03368-f004:**
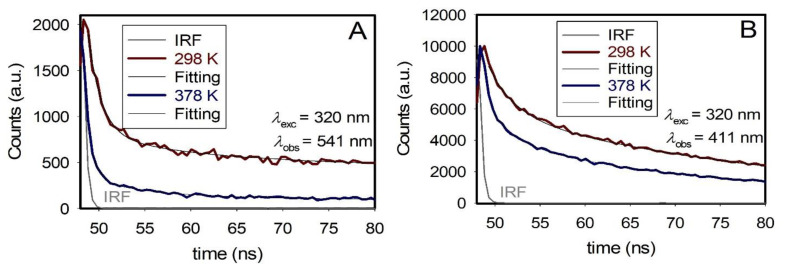
PL decays upon excitation at 320 nm monitored at the yellow band (**A**) and UV band (**B**) of ZnO–SiO_2_ solids as a function of temperatures in kelvin (K). The exact monitoring wavelengths are indicated directly on the graph.

**Table 1 nanomaterials-12-03368-t001:** Weight and atomic percentages of the SiO_2_-coated ZnO nanocrystals.

Element	Weight %	Atomic %
Zn	20.57	9.07
Si	61.17	62.73
O	15.42	27.77

**Table 2 nanomaterials-12-03368-t002:** The observed excited-state PL lifetime for different ZnO–SiO_2_ samples.

Samples	λ_obs_ (nm)	*τ*_1_ (ns)	*f*_1_%	*τ*_2_ (ns)	*f*_2_%	*τ*_3_ (ns)	*f*_3_%	*τ*_average_ (ns)	Chi-Square
298 K	411	2.7	21	20	22	49.6	57	33	1.091
298 K	541	1.4	28	12	15	111.5	57	66	1.041
378 K	411	2.5	9	16.4	27	46.8	64	35	1.002
378 K	541	0.81	52	7.03	14	67.55	34	24	1.061

The time resolution was ~90 ps and the excitation wavelength was 320 nm.

**Table 3 nanomaterials-12-03368-t003:** Amplitudes and maxima of the decay-associated spectra (DAS) for the lifetime components of ZnO–SiO_2_ solid films at different temperatures.

Temperature	*τ*/ns Excited-State Lifetimes			DAS Maximum/nm			DAS Amplitudes			% of 3
	1	2	3	1	2	3	1	2	3	
263 K	0.5	4	90	<500	<550	600	0.59	0.01	0.40	40
298 K	0.8	6	92	<500	<550	600	0.50	0.20	0.30	30
378 K	0.7	7	68	<500	<550	600	0.90	0.04	0.06	6

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
