# Peer review of "Temperature Control of Yellow Photoluminescence from SiO_2_-Coated ZnO Nanocrystals"

_nanomaterials, 2022, doi:10.3390/nano12193368_

Round 1

Reviewer 1 Report

My worst concern refers to the nature of the emitting centres and the phenomena associated with the lifetime changes. From lines 220-262, one cannot elucidate what the autors mean. In particular it is not clear the relation between the nature of the emission and mechanisms yielding a lifetime change (or no change). The occurrence of surface-based defects and their spectroscopic properties is a point that I would like to understand better through the text. The authors deal with that with a not very suitable reference [22], in a considerable different set of materials and do not provide the relationship. First they should provide more insight to have a clearer and more rigurous description of the involved emitting centres, and then provide the relation with the observed emission and excitation properties.

Author Response

We agree with the reviewer that references 22 and 29 caused some confusion, we have, thus, removed 22 and replaced 29 with a more appropriate one.

Although, the emitting canters were clearly stated in lines 47-49: “The blue and green PL of ZnO–SiO2 films have already been attributed to radiative recombination and surface defects [20].”

Yet, the following paragraph was inserted to replace the paragraph in lines 236-238; and to explain the relation between the nature of the emission and mechanisms yielding a lifetime change

Temperature control of PL from ZnO materials upon coating by SiO2 reveals for the first time the generation of the thermally controlled nonradiative channel in ZnO-based nanocomposites. In general, nonradiative recombination processes that could compete with radiative donor-acceptor recombination pair (DAP) emission from the emitting centers in solids can occur through either multiphoton emission (MPE) or energy transfer (Auger) processes. In our ZnO-SiO2 system, we conclude, for the first time, that the recombination of the electron beam excited free carriers via deep centers (DC) competes with the surface defect-assisted yellow emission. This is true because the capture process via DC is thermally activated as opposed to other nonradiative mechanisms [29].

Ref 29:

Godlewski, M.; Zakrzewski, A. Nonradiative Recombination Processes in II-VI Luminophores. II-VI, Editor, Mukesh Jain, University of Western Australia  

Reviewer 2 Report

Dear authors,

you have done a good scientific work. However, I am noticed a number of inaccuracies that need to be corrected:

Line 102: The formula for Eg does not contain Planck's constant h and the speed of light c, therefore it makes no sense to give their values.

Line 181: Figure 1 should replace by Figure 2.

Line 197: (Experimental Section) should replace by (Experimental Methods).

Line 212: (Experimental Section) should replace by (Experimental Methods).

Finally, if possible, it would be nice to show images of nanorods with significantly different lengths and diameters in Figure 1. It would beautify your article.

Author Response

Line 102: The formula for Eg does not contain Planck's constant h and the speed of light c, therefore it makes no sense to give their values.

Corrected.

Line 181: Figure 1 should replace by Figure 2.

Corrected

Line 197: (Experimental Section) should replace by (Experimental Methods).

Corrected

Line 212: (Experimental Section) should replace by (Experimental Methods).

Corrected

Finally, if possible, it would be nice to show images of nanorods with significantly different lengths and diameters in Figure 1. It would beautify your article.

Following the reviewer’s comment, we have zoomed in to 1 nm for demonstration. Although TEM images at the lowest magnification possible gave clearer nanorods, yet to distinguish the SiO2 from the ZnO, 20-5 nm magnification was the best to distinguish the core and coating. Indeed, as indicated in Figure 1, the nanorod is more evident as we zoom in from 20nm, 10nm, and finally to 5nm, however, due to particle stacking it’s less clear at higher magnification.

Reviewer 3 Report

The paper reports the preparation of ZnO-SiO2 nanocomposites and investigates their PL properties. The temperature-controlled PL spectra and time-resolved PL are carefully studied. The temperature-dependent PL is interesting. However, the manuscript is preliminary and a major revision is needed.

1. The morphology and structure of ZnO-SiO2 nanocomposites are not clear in the TEM images. It is not easy to distinguish ZnO and SiO2. The authors claim the morphology of ZnO is nanorod. However, the TEM images do not support it. The TEM image of pure ZnO should be presented. Maybe “nanorod” is not appropriate to describe the sample.

2. Maybe Fig. S4 should be placed in the main text, because it is important for understanding the PL properties of ZnO-SiO2.

2. The section of “Experimental Methods” is messy and should be rewritten. The characterizations could be put together.

3. Why tri-exponential function is used to fit the time-resolved spectra? What are the three decay processes?

4. How about the PL properties (static and time-resolved spectra) of pure ZnO? What is the role of SiO2 for the PL?

Author Response

  1. The morphology and structure of ZnO-SiO2 nanocomposites are not clear in the TEM images. It is not easy to distinguish ZnO and SiO2. The authors claim the morphology of ZnO is nanorod. However, the TEM images do not support it. The TEM image of pure ZnO should be presented. Maybe “nanorod” is not appropriate to describe the sample.

To address the reviewer’s comment, we have taken other images and confirmed the nanorod shape.

  1. Maybe Fig. S4 should be placed in the main text, because it is important for understanding the PL properties of ZnO-SiO2.

We agree with the reviewer, but we also feel Figure 3 is more important as it shows the temperature variation.

  1. The section of “Experimental Methods” is messy and should be rewritten. The characterizations could be put together.

Corrected

  1. Why tri-exponential function is used to fit the time-resolved spectra? What are the three decay processes?

This was described in lines 230-233

  1. How about the PL properties (static and time-resolved spectra) of pure ZnO? What is the role of SiO2 for the PL?

This was explained in lines 246-256

For the role of SiO2, we added

““Temperature control of PL from ZnO materials upon coating by SiO2 reveals for the first time the generation of the thermally controlled nonradiative channel in ZnO-based nanocomposites. In general, nonradiative recombination processes that could compete with radiative donor-acceptor recombination pair (DAP) emission from the emitting centers in solids can occur through either multiphoton emission (MPE) or energy transfer (Auger) processes. In our ZnO-SiO2 system, we conclude, for the first time, that the recombination of the electron beam excited free carriers via deep centers (DC) competes with the surface defect-assisted yellow emission. This is true because the capture process via DC is thermally activated as opposed to other nonradiative mechanisms [29].

Reviewer 4 Report

The paper deals with the preparation of ZnO-SiO2 nanocomposites. A wide range of methods is employed to characterize their structural and optical properties. The topic is of interest to the community dealing with nanostructured ZnO; however, it needs to be significantly improved to be accepted in Nanomaterials.

It is not clear what the state of the art is in the chemical synthesis of ZnO/SiO2 nanocomposites, why the presented chemistry was used, and what the purpose of each step in the fabrication procedure is. References should be added, reasons for using the chemicals should be given, and the novelty of the approach should be discussed.

A clear goal of the paper should be presented. Photocatalysis is mentioned as a potential application of ZnO/SiO2 nanocomposites. At the moment I do not understand how the quenching of PL with temperature would be helpful in photocatalysis. Could you discuss in more detail how exactly the field of photocatalysis would benefit from your approach/findings?

Temperature dependence of the deep-level emission from ZnO was discussed in many papers. What do you consider new in your paper?

Concerning the abstract and conclusions, the purpose is not to state that a ZnO/SiO2 nanocomposite was prepared and characterized by all available methods. The purpose is to point out the key physical/optical phenomena. The novelty of your approach/findings is not visible.

It does not make sense to have a separate subsection for each characterization technique.

Author Response

Comment 1

It is not clear what the state of the art is in the chemical synthesis of ZnO/SiO2 nanocomposites, why the presented chemistry was used, and what the purpose of each step in the fabrication procedure is. References should be added, reasons for using the chemicals should be given, and the novelty of the approach should be discussed.

Reply to Comment 1

Following the reviewer’s comments, two references were added to the synthesis part

The ZnO nanoparticles were first generated from a colloidal solution of ZnO in 250 ml ethanol before mixing them with 5.5 g of zinc acetate dehydrate (purity >98% by SigmaAldrich). The final suspension was then homogenized by heating under reflux for 1hr [21]. The distillation method was used to remove 150 ml of the solvent. After adding a new fresh volume of ethanol, 1.39 g of lithium hydroxide monohydrate (Aldrich) was added, and the new suspension was blended in an ultrasonic bath at 333 K.  The dispersed mixture for 1 hour was assumed to contain pure nanoparticles of ZnO, which was then filtered through a 0.1 mm membrane filter to remove undissolved LiOH. The final white precipitate of the ZnO nanoparticles was then heated in 100 ml of 10% of deionized water in ethanol at 333 K, centrifuged, and then washed four times with an ethanol-water mixture (19:1) to remove residual ionic compounds [22].

[21] Spanhel L.; Anderson M. A. Semiconductor Clusters in the Sol-Gel Process: Quantized Aggregation, Gelation, and Crystal Growth in Concentrated Zinc Oxide Colloids. J. Am. Chem. Soc., 1991, 113, 2826–2833.

[22] Hoyer P.; Eichberger R.; Weller H. Spectroelectrochemical Investigations of Nanocrystalline ZnO Films, Berichte Bunsenges. Für. Phys. Chem., 1993, 97, 630–635.

Comment 2

A clear goal of the paper should be presented. Photocatalysis is mentioned as a potential application of ZnO/SiO2 nanocomposites. At the moment I do not understand how the quenching of PL with temperature would be helpful in photocatalysis. Could you discuss in more detail how exactly the field of photocatalysis would benefit from your approach/findings?

Reply to Comment 2

Following the reviewer’s comments, reference 19 was modified with a more relevant one. In reference 19, the authors explicitly stated: “Temperature is usually a vital factor controlling kinetics and thermodynamics of a reaction, but it has been less investigated in photocatalysis. In this work, the effect of reaction temperature on photocatalysis was investigated in a simple process, photocatalytic degradation of Congo Red (CR) on three typical catalysts, g-C3N4, TiO2, and ZnO, to differentiate the interfacial radical generation and reaction mechanism.”

It is our assessment that trapping the charge carriers via DC, as concluded by our work, should give further insights on understating the change in photocatalytic activity of ZnO materials upon changing the temperature.  

Ref 19:

Meng, F.; Liu, Y.; Wang. J.; Tan, X.; Sun, H.; Liu, S.; Wang, S. Temperature dependent photocatalysis of g-C3N4, TiO2 and ZnO: Differences in photoactive mechanism, J. Colloid Interface Sci., 2018, 532, 321–330.

Comment 3

Temperature dependence of the deep-level emission from ZnO was discussed in many papers. What do you consider new in your paper?

Reply to Comment 3

The following paragraph was inserted to replace the paragraph in lines 236-238, and to explain our new findings: “Temperature control of PL from ZnO materials upon coating by SiO2 reveals for the first time the generation of the thermally controlled nonradiative channel in ZnO-based nanocomposites. In general, nonradiative recombination processes that could compete with radiative donor-acceptor recombination pair (DAP) emission from the emitting centers in solids can occur through either multiphoton emission (MPE) or energy transfer (Auger) processes. In our ZnO-SiO2 system, we conclude, for the first time, that the recombination of the electron beam excited free carriers via deep centers (DC) competes with the surface defect-assisted yellow emission. This is true because the capture process via DC is thermally activated as opposed to other nonradiative mechanisms [29].

Ref 29:

Godlewski, M.; Zakrzewski, A. Nonradiative Recombination Processes in II-VI Luminophores. II-VI, Editor, Mukesh Jain, University of Western Australia  

Comment 4

Concerning the abstract and conclusions, the purpose is not to state that a ZnO/SiO2 nanocomposite was prepared and characterized by all available methods. The purpose is to point out the key physical/optical phenomena. The novelty of your approach/findings is not visible.

Reply to Comment 4

We thank the reviewer for this astute comment, the following statement was added: “ Aimed at elucidating the effects of temperature on the photoluminescence from ZnO nanomaterials, the articles ….”

Comment 5

It does not make sense to have a separate subsection for each characterization technique.

Reply to Comment 5

Corrected

Round 2

Reviewer 1 Report

The modified paragraph in lines 224 and the following does not sufficiently improve insight. This point still needs better discussion for more meaningful interpretation. For instance, the authors emphasize in their answer ref. 20. How does the measure band indentify with those of the reference? (note the mismatch of the given wavelengths/energies and lifetimes, are an order of magnitude shorter)

Author Response

Reviewer 1 (second revision)

Comment 1

The modified paragraph in lines 224 and the following does not sufficiently improve insight. This point still needs better discussion for more meaningful interpretation. For instance, the authors emphasize in their answer ref. 20. How does the measure band indentify with those of the reference?

Reply to Comment 1

We agree with the reviewer that ref 20 does not support our argument and in fact, confuses the readers. We have revised the paragraph starting from line 224 to explain our new findings. We have replaced reference 20 with another one. We appreciate his comment that indeed has improved our writing.

Reference 20: Shi, J.; Chen, J.; Feng, Z.; Tao, C.; Wang, X.; Ying, P.; Li, C. Time-Resolved Photoluminescence Characteristics of Subnanometer ZnO Clusters Confined in the Micropores of Zeolites. J. Phys. Chem. B 2006, 110, 51, 25612–25618

The following paragraph/correction was added

“The DAS measurements more accurately indicate the effects of temperature on both the PL lifetime values and the contribution values of all components conducted globally at all wavelengths. It also helps us to assign each lifetime component to its DAS maximum (Table 3 and Figure S6 in the Supporting Information). It is observed that the PL lifetime values and amplitudes of the first two shorter Lifetimes 1 and 2 are not affected by temperature change (Table 3) and, therefore, belong to ZnO species which emit at 385 nm and are presumably associated with two localized exciton species [20]. However, the third component (associated with the more fluorescent ZnO/SiO2) is solely affected by temperature. The decreased PL lifetime value (Table 3) and the decreased contribution of the third lifetime Component 3 (from 40 to 6%) confirm its population transfer through a nonradiative process, see below.

The origin of UV and visible emission of ZnO has been attributed to near band edge emission and intrinsic defects such as oxygen vacancies, respectively [11]. One report suggested that the decrease in the intensity of visible emission at 298 K has resulted when surface defects are effectively passivated by the SiO2 [11]. Another research work reveals an increase in the exciton lifetime values in the presence of SiO2 [20]. In the present work, we specifically observed temperature-dependent changes in the intensity of the visible emission for the first time and attributed them to the presence of SiO2.

The results also agree with DRS measurements at different temperatures in which the bandgap at 3.3 eV belongs to ZnO [1], while the bandgap at 3.7 ev, which is sensitive to temperature, is associated with the SiO2-coated ZnO nanorods.”

Comment 2

(note the mismatch of the given wavelengths/energies and lifetimes, are an order of magnitude shorter)

Reply to Comment 2

It is true because the global fitting (Table 3) is more accurate than the non-global fitting (Table 2). Notice the error depends on the value of the lifetime as well, check Experimental Methods “The estimated experimental error was 2% for PL lifetime less than 1 ns and 20% for PL lifetime between 20 and 100 ns.”

Reviewer 3 Report

In this revised version, there still have some confusing issues:

1. The authors reply that they have provided some new images for confirming the nanorod shape. However, no new data are shown in the revised manuscript. Why not show TEM images of pure ZnO? If rod shape is not confirmed, "nanocrystals" could be used to replace "nanorods".

2. I think the PL and time-resolved spectra of pure ZnO should be given for understanding the results. 

Author Response

Reviewer 3

Comment 1

In this revised version, there still have some confusing issues:

  1. The authors reply that they have provided some new images for confirming the nanorod shape. However, no new data are shown in the revised manuscript. Why not show TEM images of pure ZnO? If rod shape is not confirmed, "nanocrystals" could be used to replace "nanorods".

Reply to Comment 1

We thank the reviewer for his concern, below TEM image clearly shows a single nanorod assembly of SiO2/ZnO. Nevertheless, we prefer to keep the TEM image at the 10 nm scale of the original manuscript, since it shows more than one nanorod.

Comment 2

  1. I think the PL and time-resolved spectra of pure ZnO should be given for understanding the results.

Reply to Comment 2

Following the reviewer’s comment, the following paragraph was added:

“The origin of UV and visible emission of ZnO has been attributed to near band edge emission and intrinsic defects such as oxygen vacancies, respectively [11]. One report suggested that the decrease in the intensity of visible emission at 298 K has resulted when surface defects are effectively passivated by the SiO2 [11]. Another research work reveals an increase in the exciton lifetime values in the presence of SiO2 [20]. In the present work, we specifically observed temperature-dependent changes in the intensity of the visible emission for the first time and attributed them to the presence of SiO2. “

Reviewer 4 Report

The authors attempted to respond to my questions and suggestions; however, my criticisms persist. I recommend that in the present form the paper is not accepted in Nanomaterials and that the authors prepare a new submission, in which they better define the goals of the study and the novelty of their approach. The present form of the manuscript is not reader-friendly and the impact on the scientific community would be low.

I recommend that the authors first discuss the optical properties of bare ZnO nanostructures and identify the origin of particular transitions in the deep-level part of the PL spectra. Then they continue with the ZnO-SiO2 composites and explain the impact of the SiO2 coating on the properties of the ZnO nanostructures. Next, they carry out advanced analysis of the optical properties, including PLE and time-resolved PL. Finally, they discuss the observed physical phenomena and confront them with the state of the art in the field.

The present form of the manuscript is sometimes hard to understand. In line number 52, the green emission is discussed,  in line 186 the authors state that two electronic species are identified (UV band and yellow band at 590 nm) even though the deep-level part of the spectrum contains at least three principal bands (deconvolution needed). Then it is not explained why the time-resolved measurements are performed at 541 nm.

In line 183 the authors state that the optical bandgap in the red region remains unchanged at 375 nm; however, 375 nm is not a red region. Moreover, the DRS spectra are not discussed in detail.

In line 266 the authors discuss the impact of the electron beam on the free carriers, but they do not mention the origin of the electron beam. According to methods, they have not used cathodoluminescence spectroscopy.

Author Response

Reviewer 4

Comment 1

The authors attempted to respond to my questions and suggestions; however, my criticisms persist. I recommend that in the present form the paper is not accepted in Nanomaterials and that the authors prepare a new submission, in which they better define the goals of the study and the novelty of their approach. The present form of the manuscript is not reader-friendly and the impact on the scientific community would be low.

I recommend that the authors first discuss the optical properties of bare ZnO nanostructures and identify the origin of particular transitions in the deep-level part of the PL spectra. Then they continue with the ZnO-SiO2 composites and explain the impact of the SiO2 coating on the properties of the ZnO nanostructures. Next, they carry out advanced analysis of the optical properties, including PLE and time-resolved PL. Finally, they discuss the observed physical phenomena and confront them with the state of the art in the field.

Reply to comment 1

We thank the reviewer for his suggestion, which certainly would improve our manuscript. We have made significant changes to the abstract, introduction, and discussion sections, as follows:

Lines 46-55

“The UV and visible PL of ZnO–SiO2 films have already been attributed to radiative recombination and surface defects [11]. Here, we analyze TRPL measurements at different temperatures to understand the photophysical role of SiO2 coating in generating nanocomposites with temperature-dependent PL aside from its role in increasing the exciton lifetime values [20]. The interplay of thermally induced nonradiative recombination transfer via deep centers (DC) of the excited-state population of the yellow emission band is confirmed. “

Lines 225-242

“The DAS measurements more accurately indicate the effects of temperature on both the PL lifetime values and the contribution values of all components conducted globally at all wavelengths. It also helps us to assign each lifetime component to its DAS maximum (Table 3 and Figure S6 in the Supporting Information). It is observed that the PL lifetime values and amplitudes of the first two shorter Lifetimes 1 and 2 are not affected by temperature change (Table 3) and, therefore, belong to ZnO species which emit at 385 nm and are presumably associated with two localized exciton species [20]. However, the third component (associated with the more fluorescent ZnO/SiO2) is solely affected by temperature. The decreased PL lifetime value (Table 3) and the decreased contribution of the third lifetime Component 3 (from 40 to 6%) confirm its population transfer through a nonradiative process, see below.

The origin of UV and visible emission of ZnO has been attributed to near band edge emission and intrinsic defects such as oxygen vacancies, respectively [11]. One report suggested that the decrease in the intensity of visible emission at 298 K has resulted when surface defects are effectively passivated by the SiO2 [11]. Another research work reveals an increase in the exciton lifetime values in the presence of SiO2 [20]. In the present work, we specifically observed temperature-dependent changes in the intensity of the visible emission for the first time and attributed them to the presence of SiO2.

The results also agree with DRS measurements at different temperatures in which the bandgap at 3.3 eV belongs to ZnO [1], while the bandgap at 3.7 ev, which is sensitive to temperature, is associated with the SiO2-coated ZnO nanorods.”

Comment 2:

The present form of the manuscript is sometimes hard to understand. In line number 52, the green emission is discussed,  in line 186 the authors state that two electronic species are identified (UV band and yellow band at 590 nm) even though the deep-level part of the spectrum contains at least three principal bands (deconvolution needed). Then it is not explained why the time-resolved measurements are performed at 541 nm.

Reply to comment 2

Corrected

The UV and visible PL of ZnO–SiO2 films have already been attributed to radiative recombination and surface defects [11]. Here, we analyze TRPL measurements at different temperatures to understand the photophysical role of SiO2 coating in generating nanocomposites with temperature-dependent PL aside from its role in increasing the exciton lifetime values [20]. The interplay of thermally induced nonradiative recombination transfer via deep centers (DC) of the excited-state population of the yellow emission band is confirmed.”

The two bands in the visible regions are overlapping at around 541 nm. We agree on the need for deconvolution, but the average lifetime value is considered here, and it was not our focus to assign each band to a specific emitting center. Both visible bands (550 and 600 nm) are attributed to intrinsic defects as reported by literature. The band at 600 nm has a temperature-dependent property, which highlights our new findings.

Comment 3:

In line 183 the authors state that the optical bandgap in the red region remains unchanged at 375 nm; however, 375 nm is not a red region. Moreover, the DRS spectra are not discussed in detail.

Reply to comment 3

To avoid confusion, we performed the following changes

Lines 182-185

By increasing the temperature of the SiO2-coated ZnO slides from 263 K to 298 K, only the UV–visible absorption edge in the blue region shifts from 335 to 355 nm, which corresponds to decreasing its optical bandgap from 3.7 to 3.5 eV (Figure 2), whereas the optical bandgap in the red region remains unchanged at 3.3 eV (375 nm).

The DRS results were discussed in lines 234-242

The origin of UV and visible emission of ZnO has been attributed to near band edge emission and intrinsic defects such as oxygen vacancies, respectively [11]. One report suggested that the decrease in the intensity of visible emission at 298 K has resulted when surface defects are effectively passivated by the SiO2 [11]. Another research work reveals an increase in the exciton lifetime values in the presence of SiO2 [20]. In the present work, we specifically observed temperature-dependent changes in the intensity of the visible emission for the first time in the presence of SiO2.

The results also agree with DRS measurements at different temperatures in which the bandgap at 3.3 eV belongs to ZnO [1], while the bandgap at 3.7 ev, which is sensitive to temperature, is associated with the SiO2-coated ZnO nanorods.

Comment 4:

In line 266 the authors discuss the impact of the electron beam on the free carriers, but they do not mention the origin of the electron beam. According to methods, they have not used cathodoluminescence spectroscopy.

Reply to comment 4

We agree with the reviewer’s comment and “free electron-excited” was removed. For the record, the cathodoluminescence spectroscopy has already been employed on ZnO nanorods by others confirming that the greenish-yellow emission is tunable with voltages. TRPL data with 3 main components (Notice the observed shorter lifetime values in the absence of SiO2 compared to our results) were collected, as well, with excitation at 375 nm and emission at 575 nm. The origin of the two main bands (UV and visible) was attributed to the near-band edge band and oxygen defects. No assignment for each visible lifetime component was made.

Ref: Conference Paper in  Materials Research Society symposia proceedings. Materials Research Society · December 2013 DOI: 10.1557/opl.2014.254

Round 3

Author Response

Thank you.

Reviewer 3 Report

There is no a large-scale TEM image of bare ZnO. Readers have no idea of the morphology, size, dispersibility of the ZnO sample, and cannot distinguish the ZnO and SiO2 domains in Fig. 1. I suggest to add a large-scale TEM image of ZnO, at least in SI.

Only one nanorod is shown, and the claim of nanorod for the ZnO sample is not solid. I suggest using "nanoscrystal" to replace "nanorod".

Author Response

We thank the reviewer for his suggestion, we have replaced “nanorod” with “nanocrystal”

The TEM image for bare ZnO has been reported already

Reviewer 4 Report

The authors have improved critical parts of the manuscript.

Author Response

Thank you.
